# 🦸 HyperLLaVA: Dynamic Visual and Language Expert Tuning for Multimodal Large Language Models

## Abstract

Recent advancements indicate that scaling up Multimodal Large Language Models (MLLMs) effectively enhances performance on downstream multimodal tasks. The prevailing MLLM paradigm, *e.g.*, LLaVA, transforms visual features into text-like tokens using a *static* vision-language mapper, thereby enabling *static* LLMs to develop the capability to comprehend visual information through visual instruction tuning. Unfortunately, the *static* paradigm shares the same parameters to underly multi-task instruction tuning, inevitably introducing the potential *task interference* or *negative transfer*, *i.e.*, where an improvement in the performance of one task reduces the performance of other tasks. In light of this, we introduce **HyperLLaVA**, which in conjunction with a dynamic visual expert and language expert, respectively adjusts the parameters of the projector and LLM layers conditioned on diverse instruction semantics, thereby minimizing the task interference. These experts are derived from HyperNetworks, which adaptively generates dynamic parameter shifts through visual and language guidance, enabling dynamic vision-language alignment and instruction tuning in two-stage training. To deeply study the multi-task interference of MLLM, we build the **Comprehensive Multimodal Task benchmark** (CMT), a comprehensive benchmark for the evaluation of multidimensional multimodal tasks. The experiments demonstrate that the superiority of the dynamic tuning paradigm for multi-task instruction following on CMT and general MLLM benchmarks. Our project is available at https://anonymous.4open.science/r/HyperLLaVA-D58E.

## 1 Introduction

The landscape of Large Language Models (LLMs) Devlin et al. (2018); Radford et al. (2018); Ouyang et al. (2022) has undergone significant evolution, highlighting their exceptional versatility in managing a wide variety of language-centric applications. To extend the capabilities of LLMs to a wider array of modal inputs, Multimodal Large Language Models (MLLMs) have garnered increasing attention Radford et al. (2021b); Li et al. (2022); Huang et al. (2023); Achiam et al. (2023); Li et al. (2023c). MLLMs are crucial for the development of flexible, general-purpose assistants, as everyday interactions encompass information from various modalities in addition to text.

Contemporary MLLMs (*e.g.*, LLaVA Liu et al. (2023c;a)) typically adhere to a two-step *static* training protocol: (i) **Vision-Language Alignment**: A *static* projector is trained by leveraging image-text pairs to synchronize visual features with the language model's word embedding space. The projector, with *static* parameters, bridges the vision and language modalities by converting visual features into visual tokens, allowing the LLM to understand visual content. (ii) **Multimodal Insturction Tuning.** Next, multimodal instruction data are employed to fine-tune the LLM, enabling it to respond to users' varied requests involving visual content. This step is crucial for enhancing the capabilities and controllability of MLLM for improving different zero-shot multimodal capabilities. Despite the critical importance of the two-step process, the projector's structure and the LLM tuning strategy remain relatively underexplored in the literature. Quantitative analyses Wang et al. (2019) indicate that a model with static parameters trained across diverse scenarios can introduce *task interference* or *negative transfer*, where excelling in one task may impede performance on another. Furthermore,

Figure 1: (a) describes the simplified version of our HyperLLaVA. (b) shows that compared to LLaVA, our method achieves superior performance across different MLLM and our `CMT` benchmarks.

an ideal MLLM should effectively comprehend a broader range of multimodal instructions and harness generalizable reasoning capabilities across various multidimensional tasks. Building on the aforementioned insights, our investigation seeks to optimize the two-stage training process in the multi-task tuning scenario, *i.e.*, aiming to simultaneously mitigate task interference and enhance the MLLM's diverse multimodal comprehension abilities.

In this paper, we propose 🦙 **HyperLLaVA** (Figure 1(a)), transitioning from *"static to dynamic tuning paradigm"* to achieve the stated objectives. The dynamic characterization benefits from a carefully designed expert module, derived from HyperNetwork Ha et al. (2017), to generate the dynamic parameters conditioned on instruction-aware semantics. Our bootstrapping philosophy is to leverage the expert to adaptively generate the strongly correlated MLLM's parameter shifts, according to the visual and language input, thereby enabling positive transfer for projector and LLM layers, respectively. By doing so, this dynamic characterization allows us to achieve the best of both worlds by adjusting the MLLM's parameters while encouraging the model to adapt to each individual multimodal instruction. Notably, in HyperLLaVA, we tailor the HyperNetwork to MLLM, incorporating input guidance-aware parameter generation and a stable learning framework through an adapter. Based on the devised expert module, HyperLLaVA is learned following the two steps: (i) In vision-language alignment, we divide the projector into static layers (original MLPs in LLaVA) and dynamic layers (visual expert), where the parameters of static layers remain fixed and the parameters of dynamic layers are dynamically generated based on visual features. The visual expert leverages HyperNetwork to assist the static projector in developing a visual-specific projector that adaptively models the visual features based on the visual guidance. Thus, the projector can deliver adaptive visual tokens to the language semantic space. (ii) For multimodal instruction tuning, we equip the LLM with a language expert, modeling dynamic parameters for LLM blocks. We regard the intermediate output of the LLM as language guidance that guides the language expert to offer an enhanced instruction-specific comprehension of the user's request. By doing so, the MLLM increases flexibility by generating unique parameters for each specific input, allowing the MLLM to capitalize on similarities between samples across tasks and avoid potential interference among different multimodal instructions.

To thoroughly investigate the issue of multi-task negative interference, we initially developed the Comprehensive Multimodal Task (`CMT`) benchmark, grounded in different interference dimensions, including multimodal processing, recognition, and comprehension. `CMT` encompasses 7 diverse multimodal tasks, including *Text-Rich Images QA*, *Spatial Inference*, *Knowledge OCR*, among others. We conducted a systematic evaluation of the proposed CMT. **The results suggest that HyperLLaVA's performance is positively correlated with the number of training task types**, while the original LLaVA demonstrated the opposite trend, highlighting the superiority of "dynamic" learning for multi-task instruction tuning. Additionally, we conducted experiments on several existing MLLM datasets, which confirmed the effectiveness and generalizability of HyperLLaVA.

## 2 METHODOLOGY

### 2.1 PROBLEM FORMULATION

The primary objective of Multimodal Large Language Models (MLLMs) is to effectively leverage the capabilities of both the pre-trained LLM and visual model. Images are considered an additional

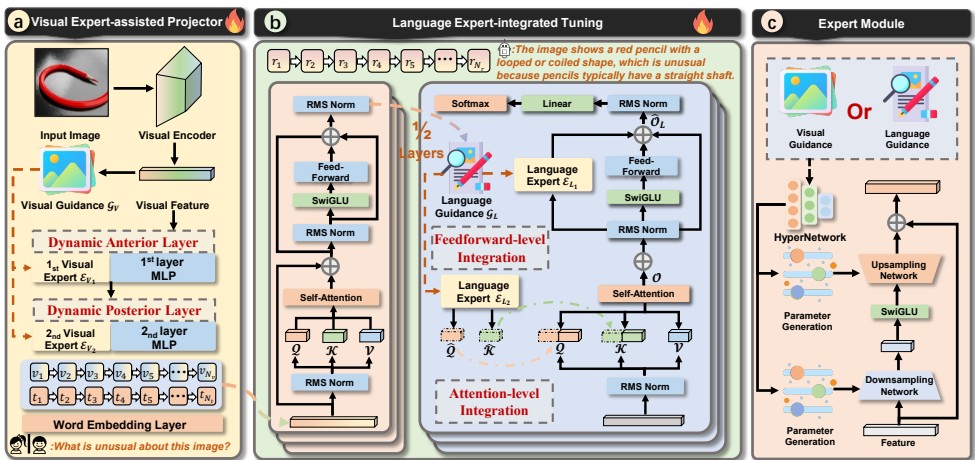

Figure 2: Overview of HyperLLaVA. (a) describes how the proposed visual expert assists the static projector in dynamically converting the image features to adaptive tokens. (b) is the language expert-integrated tuning that uses the output of the intermediate layer as language guidance to generate dynamic instruction-specific features, (c) depicts the structure of the proposed expert module.

modality input to MLLMs, making the language model a receiver of both visual and textual (instruction) tokens, and generating text responses autoregressively. The network architecture, depicted in Figure 2, comprises two steps: **Step 1** (Figure 2(a)), given an RGB image $x \in R^{H \times W \times 3}$, where $H$ and $W$ are the origin resolution. The vision encoder processes input images to obtain the visual features. Subsequently, a projector is in charge of transferring the visual features to visual tokens $\mathcal{V} = [v_1, v_2, \cdots, v_{N_v}]$ for the subsequent large language model (LLM), where $N_v$ represents the sequence length of visual tokens. **Step 2** (Figure 2(b)), we concatenate the visual tokens $\mathcal{V}$ and text tokens $\mathcal{T} = [t_1, t_2, \cdots, t_{N_t}]$, together and feed them into a LLM $\mathcal{M}_l$, then generate the language response $\mathcal{R} = [r_1, r_2, \cdots, r_{N_r}]$ by optimizing its auto-regressive training objective, where $N_t$ and $N_r$ indicate the length of text tokens and textual response, respectively. In general, the two-step learning paradigm for the MLLM model $\mathcal{M}(\cdot)$ can be described as below:

$$\underbrace{\mathcal{M}(\cdot)}_{\text{MLLM}} : \underbrace{\mathcal{M}_p((\mathcal{V}|x); \Theta_p)}_{\text{Projector}} \rightarrow \underbrace{\mathcal{M}_l((\mathcal{R}|\mathcal{V}, \mathcal{T}); \Theta_l)}_{\text{LLM}}, \quad (1)$$

where $\mathcal{M}_p(\cdot; \Theta_p)$ is the projector and $\mathcal{M}_l(\cdot; \Theta_l)$ LLM tuning with multi-modal instructions with parameters $\Theta_p$ and $\Theta_l$, respectively.

## 2.2 VISION-LANGUAGE GUIDED EXPERT MODULE

Original LLaVA's Liu et al. (2023c) projector and LLM are trained with static parameters. We argue that the static tuning paradigm may limit the flexible visual token delivery and introduce negative transfer in different downstream multi-modal tasks. Thus, we propose to equip the original's LLaVA projector and LLM with a visual expert $\mathcal{E}_V$ and a language expert $\mathcal{E}_L$: (i) the visual expert adaptively fits the projector's output according to the specific visual guidance (*e.g*, visual features); (ii) the language expert dynamically modeling the posterior blocks of LLM through anterior LLM's block output. The expert module is derived from HyperNetwork, which is a neural network that generates the parameters for another neural network. Specifically, HyperNetwork treats the parameters of the multi-layer perception (MLP) as a matrix $K^{(n)} \in R^{N_{in} \times N_{out}}$, where $N_{in}$ and $N_{out}$ represent the number of input and output neurons of the $n^{th}$ layer of MLP, respectively. $N_{in}$ and $N_{out}$ portray the structure of the MLPs together. The generation of $K^{(n)}$ can be regarded as a matrix factorization:

$$K^{(n)} = \xi(z^{(n)}; \Theta_\xi), \forall n = 1, \cdots, N_l. \quad (2)$$

During the training procedure, $\xi(\cdot; \Theta_\xi)$ is an expert module used to model MLP. $z^{(n)}$ and $\Theta_\xi$ are randomly initialized, and $z^{(n)}$ represents the learned latent vector for the $n^{th}$ layer of the MLP.

Gradients are backpropagated to both $z^{(n)}$ and $\Theta_\xi$, facilitating their update. Instead of saving $K^{(n)}$, $z^{(n)}$ and $\Theta_\xi$ will be retained.

As HyperNetwork dynamically generates a network conditioned on the input embeddings, *i.e.*, the "dynamic characterization" can be modeled by HyperNetwork. However, directly utilizing the HyperNetwork may not satisfactorily dynamic learning for MLLM for two key reasons:

- **Weak Correlation.** The original HyperNetwork learns the latent vector to generate another model's parameters. This approach lacks a strong correlation between MLLM's dynamic parameters and input multimodal instructions.
- **Unstable Optimization.** Using a HyperNetwork to generate the parameters for the projector or LLM block results in a large parameter space, *i.e.*, $D_x \times N_{in} \times N_{out}$, $D_x$ represents the input dimension of HyperNetwork. Optimizing such a vast number of parameters is challenging, and the optimization process is inherently unstable.

To this end, we carefully tailor the HyperNetwork with the following adjustments:

**Input-Parameters Correlation.** To establish the convincing correlation between MLLM's parameters and input instructions, we propose to generate the MLLM's parameters by substituting the learned latent vector $z$ with the input's embedding. Specifically, given the prior feature $f_{x^{(i)}}$ of sample $x^{(i)}$, we first develop a layer-specific encoder $E^n(\cdot)$ that encode the $f_{x^{(i)}}$ as $\mathbf{e}^{(n)}$. This vector represents the $n^{th}$ layer parameters.

$$\mathbf{e}^{(n)} = E^n(f_{x^{(i)}}), \forall n = 1, \cdots, N_l,  \tag{3}$$

where $N_l$ is the number of the modeled layers.

Then the HyperNetwork is used to convert the embedding $\mathbf{e}^{(n)}$ into parameters, *i.e.*, we input $\mathbf{e}^{(n)}$ into the following two MLP layers to generate parameters of dynamic layers.

$$K^{(n)} = \mathbf{w}^{(n)} + \mathbf{b}^{(n)} \quad s.t. \quad \mathbf{w}^{(n)} = (W_1\mathbf{e}^{(n)} + B_1)W_2 + B_2,  \tag{4}$$

where $K^{(n)}$ denotes the $n^{th}$ layer parameters of dynamic layers. Two MLP layers's weights are denoted by $W_1$ and $W_2$, respectively. $\mathbf{b}^{(n)}$, $B_1$ and $B_2$ represent the biases.

**Unstable HyperNetwork Training.** Adapters are sub-networks with small parameters that are inserted after every attention and feed-forward layer in a model Houlsby et al. (2019). The original adapter is a parameter-efficient learning approach that learns downstream tasks by updating only a small number of parameters. The adapters consist of a pair of downsampling and upsampling layers, and a residual connection. We found that using downsampling and upsampling strategies, the HyperNetwork-generated parameters can be substantially reduced.

Given the visual and language guidance $\mathcal{G}_V, \mathcal{G}_L$, the vision-language guided expert is defined as:

$$\mathcal{E}_M(x_M) = W_M^u(\text{SwiGLU}(W_M^d(x_M))) \quad s.t. \quad W_M^u, W_M^d = \mathcal{H}_M(\mathcal{G}_M), \text{where} M \in V, L  \tag{5}$$

where $M$ indicate the modality, $W_M^u, W_M^d$ respectively denote the weights for upsampling and downsampling. SwiGLU Ramachandran et al. (2017) is the activation function: Gaussian Error Linear Unit. $\mathcal{H}_M$ is the shared HyperNetwork.

## 2.3 VISUAL EXPERT-ASSISTED PROJECTOR

In this stage, our objective is to adapt the image tokens to LLM, allowing the LLM to comprehend the instances in the images. As shown in Figure 2, we divide the projector as static layers and dynamic layers. Following LLaVA1.5 Liu et al. (2023a), we employ two-layer MLPs as the static layers. To empower the projector's expression, we develop a visual expert who learns the projector shifts to model the dynamic visual tokens. Specifically, we regard the visual feature $f_V$ extracted from the visual encoder as the visual guidance $\mathcal{G}_V$, the visual expert will adaptively assist the projector that converts $\mathcal{G}_V$ to dynamic visual tokens. As commonly known, deep neural networks encode visual features with increasing abstraction, generally, becoming finer as we progress over levels. Given two-layer MLPs, we introduce two selectable configurations for dynamic vision-language alignment: *dynamic anterior layer* and *dynamic posterior layer*.

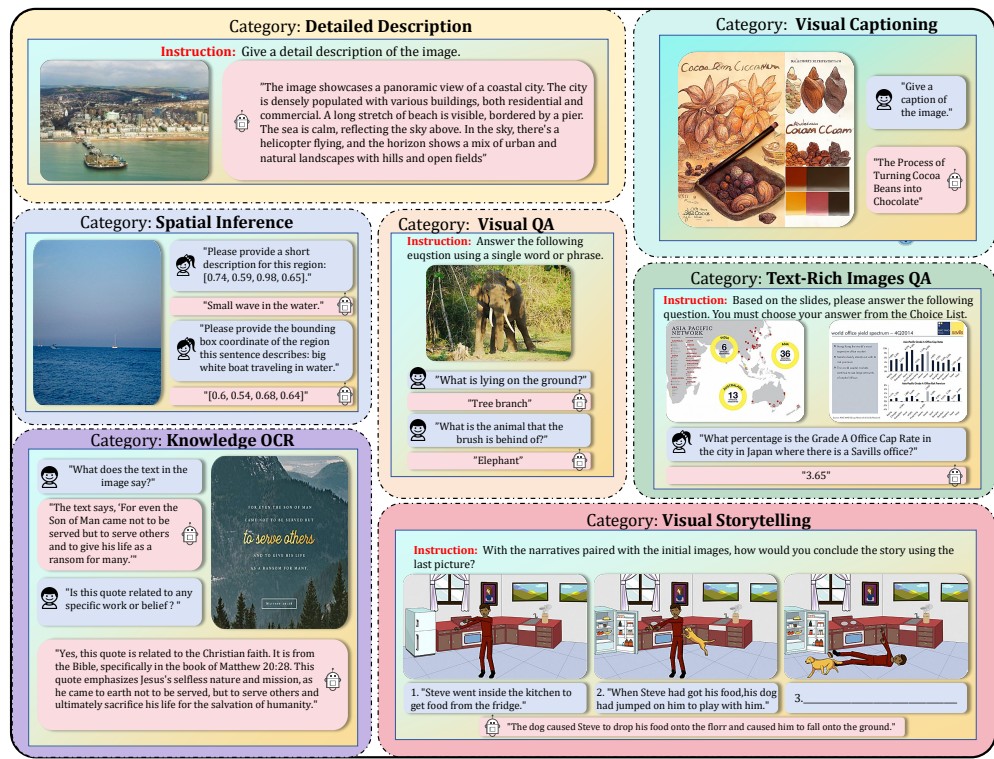

Figure 3: Demonstrations and task taxonomy of the proposed `CMT` benchmark.

**Dynamic Anterior Layer.** Taking the visual guidance $\mathcal{G}_V$ as input to 1st layer MLP and visual expert $\mathcal{E}_{V_1}(\cdot)$, we then concatenate their output to 2nd layer MLP. By doing so, the adaptive visual tokens can be obtained as $\mathcal{V} = Linear_2(Linear_1(\mathcal{G}_V) + \mathcal{E}_{V_1}(\mathcal{G}_V))$.

**Dynamic Posterior Layer.** Given the hidden representation of the 1st layer MLP for modeling the visual guidance $\mathcal{G}_V$, we input the this representation to 2nd layer MLP and visual expert $\mathcal{E}_{V_2}(\cdot)$. The dynamic modeled visual tokens can be represented as $\mathcal{V} = Linear_2(Linear_1(\mathcal{G}_V)) + \mathcal{E}_{V_2}(\mathcal{L}_1(\mathcal{G}_V))$.

These visual experts learn to adjust the projector shift to adapt visual information, modeling dynamic visual tokens and thus enhancing the projector's expressiveness for downstream tasks.

## 2.4 LANGUAGE EXPERT-INTEGRATED TUNING

In this stage, LLM is adjusted to become an MLLM with multi-modal understanding. We use more complex instructions to achieve a stronger multi-modal understanding. Previous studies have shown that features provided by the intermediate layer may suffice to preliminarily understand the given input samples Xin et al. (2020)and can serve as guidance hints to improve training Romero et al. (2014). Thus, generating guidance in the intermediate LLM layer allows the model to form a preliminary understanding of the given instruction. Therefore, we regard the output of the intermediate LLM layer as language guidance that generates adaptive instruction-specific features that enhance the generation accuracy. Taking the multimodal instruction as input to the language decoder, we then extract the hidden representation of the last input token $h^{\frac{L}{2}}$ at $\frac{L}{2}$-th layer, which can fully perceive the whole multimodal context during the $\frac{L}{2}$ layers and contains comprehensive instruction-aware semantics. In our situation, we regard the $h^{\frac{L}{2}}$ as the language guidance $\mathcal{G}_L$ and propose two alternative strategies of language expert tuning: *attention-level integration* and *feedforward-level integration*.

**Attention-level Integration.** The first language expert integration strategy is to modify the inputs of the MSA layers with instruction-specific prompts. We split the prompt into two language sub-prompts

$\hat{\mathcal{K}}$ and $\hat{\mathcal{Q}}$ and prepend them to the key and value vectors respectively. We denote the query, key and value for the multi-head self-attention (MSA) layer as:

$$\mathcal{O} = \text{MSA}([\hat{\mathcal{Q}}, \mathcal{Q}], [\hat{\mathcal{K}}, \mathcal{K}], \mathcal{V}), \quad s.t. \quad \hat{\mathcal{Q}} = \mathcal{E}_{L_1}(\mathcal{Q})^\top W^Q, \hat{\mathcal{K}} = \mathcal{E}_{L_1}(\mathcal{K})^\top W^K \tag{6}$$

where $W^Q$ and $W^K$ are the trainable weight matrice, $\mathcal{E}_{L_1}$ is the language expert.

**Feedforward-level Integration.** Another integration approach is to add extra language expert knowledge to the feedforward layer. We use the language expert $\mathcal{E}_{L_2}$ to generate the complementary information, which is integrated into the feedforward layer. The instruction-specific representation can be calculated as below:

$$\hat{\mathcal{O}}_L = \mathcal{O}_L + \text{RMS}(\mathcal{O}) + \text{FFN}(\text{SwiGLU}(\text{RMS}(\mathcal{O}))) \quad s.t. \quad \mathcal{O}_L = \mathcal{E}_{L_2}(\text{RMS}(\mathcal{O})). \tag{7}$$

Such language expert-integrated tuning enables the MLLM to measure the similarities between different multimodal instructions and thus avoid potential multi-task interference.

## 3 EXPERIMENTS

### 3.1 CMT BENCHMARK.

To thoroughly investigate the issue of multi-task negative interference and comprehensively benchmark the diverse multimodal instruction following ability, we extensively gather and annotate a wide variety of multimodal datasets from different fields and scenarios. As illustrated in Figure 3, CMT has diverse forms of complex instructions and a vast range of instruction-following scenarios, covering **7 tasks across 22 scenarios**, including **Visual QA** (VQA), **Visual Captioning** (VC), **Spatial Inference** (SI), **Detailed Description** (DD), **Visual Storytelling** (VS), **Knowledge OCR** (KOCR), **Text-Rich Images QA** (TQA). The tasks are selected that considered five interference dimensions: (i) *Single and Multiple Image Processing* → Visual Captioning and Visual Storytelling; (ii) *Pure Vision and Multimodal Information* → Visual QA and Text-Rich Images QA; (iii) *Visually Global and Local Details Understanding* → Detailed Description and Spatial Inference; (iv) *Visual and Textual Recognition in Images* → Spatial Inference and Knowledge OCR; (v) *Brief and Detailed Textual Understanding* → Visual Captioning and Detailed Description. All task instances are transformed into a unified instruction-response form for zero-shot evaluation, including {Task_Instruction}, {Task_Instance} and {Response}. In total, CMT includes 505,405 multi-round instruction-response pairs conversations for training and randomly selected 1,149 instruction-response pairs for evaluation. Please refer to Appendix 6 for more details of the developed CMT benchmark.

### 3.2 DATASET AND SETTING

**Benchmark Datasets.** We evaluate our proposed HyperLLaVA on five VQA datasets: VQA-v2 Goyal et al. (2017b); GQA Hudson & Manning (2019b); VizWiz Gurari et al. (2018); SQA$^I$: ScienceQA-IMG Lu et al. (2022); VQA$^T$ Singh et al. (2019a): TextVQA and seven Benchmark Toolkits: POPE Li et al. (2023e); MME Fu et al. (2023b); MMB: MMBench Liu et al. (2023d); MMB$^{CN}$: MMBench-Chinese Liu et al. (2023d); SEED: SEED-Bench Li et al. (2023b); LLaVA$^W$: LLaVA-Bench(In-the-Wild) Liu et al. (2023c); MM-Vet Yu et al. (2023).

**Implementation Details.** Our 7b model version takes approximately 18 hours to train on 8×A800 machine, while the 13b model version takes about 18.5 hours to train on 16×A800 machine. In the training of the HyperLLaVA, we utilize the ADAMW Loshchilov & Hutter (2017) optimizer, adapting hyperparameters to cater to the specific requirements of each phase. For the feature alignment stage, parameters are set as $B = 32$, $Lr = 0.001$, while for the multimodal instruction tuning stage, we adjust the parameters to $B = 16$, $Lr = 0.00002$. Additional details can be found in Appendix 7.1, maintaining consistency with LLaVA-1.5.

**Comparison of Methods.** We compare HyperLLaVA with previous SOTA approaches for quantifying the efficacy. We choose BLIP-2 Li et al. (2023d), InstructBLIP Dai et al. (2023a) based on Vicuna-7B, InstructBLIP Dai et al. (2023a) based on Vicuna-13B, Shikra Chen et al. (2023), IDEFICS-9B Laurençon et al. (2023), IDEFICS-80B Laurençon et al. (2023), Qwen-VL Bai et al. (2023), Qwen-VL-Chat Bai et al. (2023) and LLaVA-1.5 Liu et al. (2023a). More details refer to 7.2.

Table 1: **Comparison with SoTA methods on 12 benchmarks. For making a fair comparison, we use the LLaVA's data to train our model.** Res, PT, IT indicate input image resolution, the number of samples in the pretraining and instruction tuning stage, respectively. We color each row as the `best` and `second best`. Improvement. ↑ indicates performance improvement compared with LLaVA-7B and LLaVA-13B.

| Method | LLM | Res. | PT | IT | VQA Datasets | | | | | Benchmark Toolkits | | | | | | |
|---|---|---|---|---|---|---|---|---|---|---|---|---|---|---|---|---|
| | | | | | VQA$^{v2}$ | GQA | VizWiz | SQA$^I$ | VQA$^T$ | POPE | MME | MMB | MMB$^{CN}$ | SEED | LLaVA$^W$ | MM-Vet |
| InstructBLIP Dai et al. (2023a) | Vicuna-7B | 224 | 129M | 1.2M | - | 49.2 | 34.5 | 60.5 | 50.1 | - | - | 36 | 23.7 | 53.4 | 60.9 | 26.2 |
| IDEFICS-9B Laurençon et al. (2023) | LLama-7B | 224 | 353M | 1M | 50.9 | 38.4 | 35.5 | - | 25.9 | - | - | 48.2 | 25.2 | - | - | - |
| Qwen-VL Bai et al. (2023) | Qwen-7B | 448 | 1.4B | 50M | 78.8 | 59.3 | 35.2 | 67.1 | 63.8 | - | - | 38.2 | 7.4 | 56.3 | - | - |
| Qwen-VL-Chat Bai et al. (2023) | Qwen-7B | 448 | 1.4B | 50M | 78.2 | 57.5 | 38.9 | 68.2 | 61.5 | - | 1487.5 | 60.6 | 56.7 | 58.2 | - | - |
| LLaVA-1.5 Liu et al. (2023a) | Vicuna-7B | 336 | 558K | 665K | 78.5 | 62.0 | 50.0 | 66.8 | 58.2 | 85.9 | 1474.0 | 64.3 | 58.3 | 58.6 | 63.4 | 30.5 |
| **HyperLLaVA** (Ours) | Vicuna-7B | 336 | 558K | 665K | 79.1 | 62.7 | 51.9 | 70.4 | 58.5 | 86.3 | 1481.2 | 65.9 | 60.6 | 61.4 | 64.0 | 31.0 |
| **Improvement.** ↑ | - | - | - | - | +0.6 | +0.7 | +1.9 | +3.6 | +0.3 | +0.4 | +7.2 | +1.6 | +2.3 | +2.8 | +0.6 | +0.5 |
| BLIP-2 Li et al. (2023d) | Vicuna-13B | 224 | 129M | - | 41.0 | 41 | 19.6 | 61 | 42.5 | 85.3 | 1293.8 | - | - | 46.4 | 38.1 | 22.4 |
| InstructBLIP Dai et al. (2023a) | Vicuna-13B | 224 | 129M | 1.2M | - | 49.5 | 33.4 | 63.1 | 50.7 | 78.9 | 1212.8 | - | - | 58.2 | - | 25.6 |
| Shikra Chen et al. (2023) | Vicuna-13B | 224 | 600K | 5.5M | 77.4 | - | - | - | - | - | 58.8 | - | - | - | - | - |
| LLaVA-1.5 Liu et al. (2023a) | Vicuna-13B | 336 | 558K | 665K | 80.0 | 63.3 | 53.6 | 71.6 | 61.3 | 85.9 | 1531.3 | 67.7 | 63.6 | 61.6 | 70.7 | 35.4 |
| **HyperLLaVA** (Ours) | Vicuna-13B | 336 | 558K | 665K | 80.1 | 63.8 | 54.6 | 73.8 | 61.1 | 86.4 | 1571.1 | 69.0 | 63.0 | 62.9 | 70.9 | 36.6 |
| **Improvement.** ↑ | - | - | - | - | +0.1 | +0.5 | +1.0 | +2.2 | - | +0.5 | +39.8 | +1.3 | - | +1.3 | +0.2 | +1.2 |

## 3.3 OVERALL PERFORMANCE

**Existing Benchmarks.** We benchmark HyperLLaVA on a wide range of academic benchmarks, including 5 `VQA datasets` and 7 `Benchmark Toolkits` in Table 1. In general, irrespective of the different benchmarks, HyperLLaVA achieves the best performance on almost all the multimodal scenarios across both datasets. Besides, compared to LLaVA, we show that HyperLLaVA achieves the best performance across **12 out of 12 benchmarks (7B version) and 10 out of 12 benchmarks (13B version)**. Such results benefit from the carefully designed dynamic visual and language expert, which empowers the static projector and LLM to facilitate general multimodal tasks.

**CMT Benchmark.** To further measure the multimodal understanding capability, we conduct a comprehensive evaluation of our HyperLLaVA and the recent advanced MLLMs on the proposed `CMT` benchmark, which reveals several key findings: 1) **HyperLLaVA consistently outperforms existing models by a large margin across all categories**, which demonstrates stronger generalizability in following multimodal instructions with different types. 2) Despite existing vision-language models have demonstrated comparable performance in following general multimodal instructions (*e.g.*, Visual QA and Visual Captioning), their competence seems to falter when simultaneously dealing with the complex multimodal instructions (*e.g.*, Spatial Inference and Knowledge OCR). Among these widely varying multimodal tasks, this is perceived as a deficiency in multi-task interference, which may introduce the negative transfer, thus attributing the performance discrepancy. In contrast, the proposed visual and language experts can adapt MLLM's parameters conditioned for every instruction at two stages, alleviating the potential interference and improving multimodal comprehension across different tasks. 3) The original LLaVA exhibits performance degradation when scaling up the LLM size, however, our model shows consistent performance improvement for all tasks, indicating the suitability and stability for different vision-language instruction understanding.

## 3.4 IN-DEPTH ANALYSIS

We further validate the effectiveness of the HyperLLaVA-7B through the experiments on VizWiz, SQA$^I$, MMB, SEED, Visual QA (VQA) and Spatial Inference (SI) on `CMT` benchmark+.

**Task Interference Analysis.** We systematically detail the explicit task interference in Figure 4 (a) and (b), which display the experimental outcomes from training with combinations of different task data for the Visual QA and Spatial Inference tasks. Interestingly, LLaVA achieves higher or comparable performance to our proposed method when trained on single-task data. However, the results presented in the figure also reveal that LLaVA obtains significant performance degradation as the number of training task types increases, implying the limitations of LLaVA's "static" learning in the multi-task setting. In contrast, HyperLLaVA exhibits consistent performance enhancements across the two tasks as the number of training task types increases. Our intuition is that the "dynamic" visual and language expert modules effectively capture domain-specific knowledge by adaptively adjusting the MLLM's parameters, while the "static" component learns general knowledge across diverse multimodal tasks. Consequently, as the number of training tasks increases, the static part effectively enhances general knowledge, while the dynamic component mitigates potential interference, enabling positive transfer

Table 2: **Evaluation on each task category of developed CMT benchmark.** [†] indicates the zero-shot evaluation of the model. Notably, LLaVA and HyperLLaVA were both trained using the CMT data.

| Method | Visual QA | Visual Captioning | Spatial Inference | Detailed Description | Visual Storytelling | Knowledge OCR | Text-Rich Images QA |
|---|---|---|---|---|---|---|---|
| BLIP-2[†] Li et al. (2023d) | 8.4 | 8.7 | 0.0 | 17.4 | 8.9 | 21.0 | 16.0 |
| InstructBLIP[†] Dai et al. (2023a) | 35.2 | 7.1 | 0.0 | 2.7 | 10.5 | 20.1 | 17.3 |
| MiniGPT-4[†] (Zhu et al., 2023) | 0.0 | 17.4 | 0.0 | 29.0 | 9.2 | 17.9 | 17.1 |
| mPLUG-Owl[†] Ye et al. (2023b) | 71.0 | 15.0 | 9.9 | 30.3 | 9.7 | 31.3 | 14.1 |
| Otter[†] Li et al. (2023a) | 24.1 | 11.4 | 0.0 | 26.1 | 14.0 | 21.0 | 22.1 |
| Qwen-VL-Chat[†] Bai et al. (2023) | 53.1 | 13.0 | 13.1 | 21.4 | 13.7 | 31.0 | 20.1 |
| LLaVA-7B Liu et al. (2023a) | 77.5 | 15.3 | 32.7 | 31.2 | 10.9 | 43.5 | 29.0 |
| **HyperLLaVA-7B** | **79.0** | **21.3** | **36.9** | **32.2** | **15.2** | **46.3** | **30.1** |
| LLaVA-13B Liu et al. (2023a) | 77.8 | 15.0 | 37.5 | 32.0 | 11.6 | 48.2 | 31.9 |
| **HyperLLaVA-13B** | **79.6** | **21.6** | **39.0** | **35.9** | **15.2** | **52.1** | **32.8** |

Table 3: **Three alternatives for dynamic vision-language alignment.** $\mathcal{E}_{V_1}$ and $\mathcal{E}_{V_2}$ denote the visual expert for first and second MLP layer.

| Methods | VQA Datasets | | Benchmark Toolkits | | CMT Benchmark | |
|---|---|---|---|---|---|---|
| | VizWiz | SQA$^I$ | MMB | SEED | VQA | SI |
| w/o $\mathcal{E}_V$ | 50.3 | 70.4 | 65.9 | 61.0 | 77.2 | 33.0 |
| $\mathcal{E}_{V_2}$ | 51.4 | 70.9 | 64.7 | 61.0 | 78.6 | 35.6 |
| $\mathcal{E}_{V_1}$&$\mathcal{E}_{V_2}$ | 48.2 | 70.6 | 63.3 | 58.0 | 78.2 | 36.1 |
| **$\mathcal{E}_{V_1}$** | **51.9** | **70.4** | **65.9** | **61.4** | **79.0** | **36.9** |

Table 4: **Different language expert tuning strategies.** **ATT** and **FFN** denote the attention-level and feedforward-level integration.

| Methods | VQA Datasets | | Benchmark Toolkits | | CMT Benchmark | |
|---|---|---|---|---|---|---|
| | VizWiz | SQA$^I$ | MMB | SEED | VQA | SI |
| w/o $\mathcal{E}_L$ | 51.1 | 70.2 | 65.7 | 60.8 | 77.7 | 34.2 |
| ATT | 45.4 | 70.2 | 66.2 | 61.5 | 78.7 | 35.3 |
| ATT&FFN | 45.5 | 70.3 | 66.5 | 61.3 | 77.3 | 35.5 |
| **FFN** | **51.9** | **70.4** | **65.9** | **61.4** | **79.0** | **36.9** |

across projector and LLM layers in a multi-task learning scenario. This showcases HyperLLaVA's suitability and stability for diverse vision-language instruction comprehension.

**Dynamic Characterization Visualization.** We investigate the dynamic characterizations of the visual expert. Specifically, we have randomly selected 70 cases (10 cases per task) from the constructed CMT benchmark and visualized the parameters of visual and language experts using t-SNE embeddings Van der Maaten & Hinton (2008) in Figure 4(c) and (d). This visualization demonstrates the dynamic characterization of the generated parameter, *e.g*, the sample distribution is discrete in the projector and LLM. Such dynamic characterization enables the MLLM to leverage the best of both worlds, adjusting the limited MLLM parameters and encouraging the model to adapt to individual multimodal instructions, consequently alleviating the multi-task interference.

**Effectiveness of Each Component.** We investigate the effectiveness of each component in Table 3 and 4. On the one hand, Table 3 builds the insights on the visual expert-assisted projector in HyperLLaVA. According to our observation, using one visual expert to access the dynamic projection yields the best results (Row 4). Besides, the other two plans (Row 2 and Row 3) also obtained comparable results, indicating the effectiveness of dynamic vision-language projection. On the other hand, Table 3 shows the different language expert integration strategies. Comparing ATT and FFN, FFN (Row 4) shows a stable performance for all tasks, while utilizing ATT (Row 2 and Row 3) results in noticeable performance degradation on VizWiz benchmark. Our intuition is that the attention-level brings more parameter computation at all LLM blocks, and thus hurts the stability. Table 3 (Row 1) and 4 (Row 1) also suggest that the improvement of using each expert module alone is distinguishable. Combining all the components, our HyperLLaVA exhibits steady improvement over the baselines.

**Analysis of Language Expert Integration for Different Blocks.** To deeply analyze the effectiveness of language experts, we study the language expert integration for different blocks in Table 7, including anterior 16 blocks (before 1/2 LLM layers), all 32 blocks (all LLM layers) and posterior 16 blocks (after 1/2 LMM layers). Generally speaking, leveraging the language expert integration for the posterior 16 blocks obtained almost the best performance. Besides, Row 2 and Row 3 utilize the initial language input as language guidance, obtaining suboptimal results compared with language expert integration for the posterior 16 blocks. Our intuition is that the language guidance might not have gathered sufficient contextual information for subsequent dynamic LLM layer modeling.

**Analysis on the Inserted Blocks for Language Guidance.** We investigate the impact of inserting language guidance into different layers of LLMs. We report the evaluation score of VisWiz, MMB

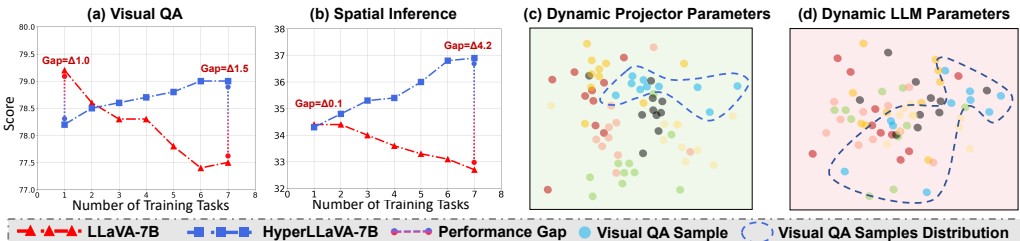

Figure 4: **Deep analysis of HyperLLaVA**. (a) and (b) report the results based on the combined training data of different tasks on `CMT` benchmark. (c) and (d) respectively visualize the dynamic parameters in the projector and LLM by using t-SNE Van der Maaten & Hinton (2008).

**Table 5: Zero-shot object hallucination evaluation results on POPE dataset.** "Yes" indicates the proportion of positive responses to the given question.

| Method | LLM | Activated | Adersarial | | | Popular | | | Random | | |
|---|---|---|---|---|---|---|---|---|---|---|---|
| | | | Acc | F1-Score | Yes | Acc | F1-Score | Yes | Acc | F1-Score | Yes |
| mPLUG-Owl Ye et al. (2023a) | LLaMA-7B | 6.7B | 82.4 | 81.6 | 45.2 | 85.5 | 84.3 | 42.1 | 86.3 | 85.3 | 42.3 |
| MM-GPT Gong et al. (2023) | LLaMA-7B | 6.7B | 50.0 | 66.7 | 100.0 | 50.0 | 66.7 | 100.0 | 50.0 | 66.7 | 100.0 |
| LLaVA-1.5 Liu et al. (2023a) | Vicuna-7B | 7B | 85.1 | 84.2 | 44.0 | 87.2 | 86.1 | 41.9 | 88.3 | 87.3 | 41.9 |
| **HyperLLaVA** | Vicuna-7B | 7B | **85.6** | **84.7** | **44.1** | **87.3** | **86.2** | **42.4** | **88.9** | **87.9** | **42.1** |

and VQA on `CMT` in Figure 5 (a), (b) and (c). We observe that the performance is low when we insert language guidance too early (*i.e.*, 4, 8) as the model might not have gathered sufficient contextual information to generate effective guidance. Meanwhile, inserting language guidance too late (*i.e.*, 24, 28) degenerates the performance. We speculate this is due to the generated guidance being too concentrated and there not being enough layers to integrate the language-aware details.

**Analysis of Expert's Structure.** We systematically present the explicit benefits from the carefully designed expert's structure in Table 6. Simply using HyperNetwork performs worse, demonstrating the unstable optimization with numerous parameters. The adapter-based HyperNetwork structure surpasses MLP across all datasets, primarily because the generated MLP is no longer a lightweight network to optimize, resulting in unstable performance. Compared with HyperNetwork+Adapter (Row 3 *vs* Row 5), our proposed vision-language guided expert structure achieved the best performance. These results align with our assumption that the original HyperNetwork lacks a strong correlation between input and parameter generation. Our method enables the exploitation of similarities between samples across datasets and avoids potential interference among different instructions.

**Effect of Dimension of Expert Input and Downsampling.** Figure 5 (d) and (e) empirically provide an appropriate dimension of input and downsampling, *i.e*, 128 and 64, respectively, either increasing or decreasing this value results in a performance decay. According to our analysis, a bigger dimension may result in an unstable HyperNetwork optimization, and a smaller value contains less language-guided information for dynamic learning, thus yielding performance decay.

**Object Hallucination Evaluation.** We adopt the evaluation pipeline of POPE Li et al. (2023e), a polling-based query method, to evaluate object hallucination in HyperLLaVA. The results are presented in Table 5, where HyperLLaVA exhibits the best performance, indicating that HyperLLaVA tends to generate objects consistent with the given image. Additionally, we observe that the "yes" ratio of HyperLLaVA remains relatively balanced, indicating that our model is capable of providing accurate feedback based on the questions.

**Effect with Stronger LLM.** To access the LLM generalizability of the proposed method, we have conducted experiments using LLaVA-1.5 training data combined with the more powerful LLM (LLaMA3-8B) utilized by LLaVA-1.6[1], as detailed in Table 13. These experiments demonstrate that HyperLLaVA significantly outperforms the LLaVA 1.6 variant across all tasks, showcasing superior generalizability in processing diverse multimodal instructions.

**MMMU Benchmark Results.** MMMU Yue et al. (2024) is a benchmark for evaluating MLLMs across multiple disciplines, which serves as an alternative for diverse task learning of MLLMs. Thus,

---

[1]*Due to LLaVA-1.6 has not yet fully open-sourced, we only replace the LLaMA2-7B to LLaMA3-8B.*

Figure 5: **Analysis of HyperLLaVA's hyperparameters**. (a)(b)(c) depicts the effect of selected blocks for language guidance. (d) and (e) demonstrates the performance on different benchmarks with respect to the input and downsampling dimensions of the designed expert module.

Table 6: **Deep analysis of expert structure.**

| Methods | VQA Datasets | | Benchmark Toolkits | | CMT Benchmark | |
|---|---|---|---|---|---|---|
| | VizWiz | SQA[1] | MMB | SEED | VQA | SI |
| Adapter | 50.7 | 69.4 | 63.9 | 56.9 | 73.4 | 32.8 |
| HyperNetwork | 36.5 | 52.6 | 51.1 | 48.8 | 70.1 | 31.3 |
| +Adapter | 51.6 | 69.9 | 65.5 | 60.8 | 75.9 | 33.6 |
| +MLP | 51.0 | 68.8 | 64.1 | 59.7 | 74.3 | 32.9 |
| **Ours** | **51.9** | **70.4** | **65.9** | **61.4** | **79.0** | **36.9** |

Table 7: **Analysis of language expert integration for different LLM layers.** $L$ indicates the number of the LLM blocks.

| Methods | VQA Datasets | | Benchmark Toolkits | | CMT Benchmark | |
|---|---|---|---|---|---|---|
| | VizWiz | SQA[1] | MMB | SEED | VQA | SI |
| Anterior $\frac{L}{2}$ Blocks | 49.3 | 69.4 | 65.0 | 59.8 | 78.2 | 35.3 |
| All Blocks | 47.8 | 69.5 | 66.1 | 59.8 | 78.0 | 35.5 |
| **Posterior $\frac{L}{2}$ Blocks** | **51.9** | **70.4** | **65.9** | **61.4** | **79.0** | **36.9** |

We conduct additional experiments to explore the other multi-modal understanding capabilities of HyperLLaVA. As shown in Table 11 (in Appendix), the results we find that HyperLLaVA notably surpasses LLaVA-1.5 on all the different tasks. The observations further reveal the superiority of HyperLLaVA, which can effectively address the negative transfer in multi-task learning.

**Human Evaluation.** We further conduct a human evaluation on the OwlEval benchmark Ye et al. (2023b), which contains 82 open-ended questions including advertisement and poem creation, diagram and flowchart comprehension, and teaching, *etc.* Specifically, we recruit 8 well-educated people to rank the randomly shuffled responses from MiniGPT-4, mPLUG-Owl, OpenFlamingo, InstructBLIP and LLaVA. The scores range from 1 to 5 (5 means best) and are allowed to be equal for comparable instances. As shown in Figure 6, HyperLLaVA also demonstrates better open-ended language generation ability in various practical cases.

## 4 RELATED WORK

**Multimodal Large Language Models (MLLMs).** MLLMs leverage the power of LLMs, mitigating extra computational cost and enhancing the efficacy of multimodal pre-training Zhang et al. (2024), to bridge the gap between textual and multimodal data. Follow-up works of LLaVA (Liu et al., 2023b), MiniGPT-4 (Zhu et al., 2023), InstructBLIP (Dai et al., 2023b), Qwen-VL-Chat Bai et al. (2023), Flamingo Alayrac et al. (2022b), Otter Li et al. (2023a), mPLUG-Owl (Ye et al., 2023b) propose to fine-tune MLLMs with multimodal instructions. To effectively benchmark the recent progress in MLLMs, concurrent works of LVLM-eHub (Xu et al., 2023) and MME Benchmark (Fu et al., 2023a) are proposed, while they mainly focus on instructions that only involve a single image with limited instruction diversity. However, most of the pieces of literature focus on scaling up the pretraining data, instruction-following data, visual encoders or language models to facilitate multimodal understanding. How to alleviate the multi-task interference of MLLMs remains relatively underexplored. Thus, we propose HyperLLaVA, addressing the task interference based on the novel dynamic tuning strategy, yielding an improved understanding of diverse multimodal instructions.

## 5 CONCLUSION

Building upon HyperLLaVA's innovative dynamic tuning strategy, our work paves the way for groundbreaking advancements in multimodal learning systems. By adaptively tuning both projector and LLM parameters, and integrating dynamical visual and language experts, we not only surpass the performance benchmarks set by LLaVA but also introduce a comprehensive multimodal task benchmark. This approach offers a new horizon for enhancing multimodal task performances through personalized, dynamic adjustments. Future research could further explore the scalability of dynamic tuning mechanisms, potentially unlocking new avenues for understanding multimodal instructions.

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

This is the Appendix for "HyperLLaVA: Dynamic Visual and Language Expert Tuning for Multimodal Large Language Models". Table 8 summarizes the abbreviations and the symbols used in the main paper.

**Table 8:** Abbreviations and symbols used in the main paper.

| Abbreviation/Symbol | Meaning |
|:---:|:---:|
| | *Abbreviation* |
| LLMs | Large Language Models |
| MLLMs | Multimodal Large Language Models |
| CMT | Comprehensive Multimodal Tasks |
| MLP | Multi-Layer Perception |
| FC | Fully-Connected |
| MSA | Multi-Head Self-Attention |
| | *Symbol in Algorithm* |
| $\mathcal{V}$ | Visual Token Sequence |
| $\mathcal{T}$ | Text Token Sequence |
| $\mathcal{R}$ | Textual Response Token Sequence |
| $\mathcal{M}_v$ | VIT Model |
| $\mathcal{M}_p$ | Projector Model |
| $\mathcal{M}_l$ | LLM |
| $\mathcal{M}$ | MLLM |
| $K$ | Dynamic MLP Matrix |
| $\xi$ | Expert Module |
| $z$ | Learned Latent Vector |
| $E$ | Layer-Specific Encoder |
| $e$ | Layer-Specific Feature Embedding |
| $M$ | Modality Type |
| $\mathcal{G}$ | Guidance |
| $\mathcal{H}$ | HyperNetwork |
| $\mathcal{E}$ | Expert |
| $\hat{\mathcal{Q}}$ | Query Sub-Prompt |
| $\hat{\mathcal{K}}$ | Key Sub-Prompt |

This Appendix is organized as follows:

- Section 6 provides the detailed information of the proposed CMT benchmark.
- Section 7 reports more experimental settings of baselines, implementation details and training process of HyperLLaVA.
- Section 8 shows the additional experiments to verify the effectiveness of HyperLLaVA.
- Section 9 lists the broader impact and limitations of this paper.

## 6 CMT BENCHMARK

The majority of the 12 benchmarks assessed in Table 1 are primarily centered on a specific task/domain (*e.g.*, Visual Question Answering (VQA)) or straightforward reasoning tasks (MME Benchmark). We contend that these benchmarks may not effectively evaluate the nuanced interplay between different tasks. Therefore, we developed the CMT benchmark, encompasses five interference dimensions among various tasks, serving as a fundamental basis for investigating task interference.

**Data format.** All task instances are transformed into a unified instruction-response form for zero-shot evaluation. Formally, each instance in CMT consists of the following components:

- `Task_Instruction`: provides a complete natural language definition of a given task, including the input/output format and the task objective.

Table 9: Detailed statistics of `CMT` benchmark.

| | Tasks | Images | Instructions | Avg. Images / Instruction | Avg. Words / Instruction |
|---|---|---|---|---|---|
| `CMT` for training | 7 | 772,867 | 505,405 | 1.53 | 28.27 |
| `CMT` for evaluation | 7 | 2,01 | 1,149 | 1.74 | 32.72 |

- `Task_Instance`: is a concrete instance of a given task that consists of demonstrative image-text sequential context (*e.g.*, visually-rich textbooks, specific questions about the context).

- `Response`: represents the target output in natural language for a given task instruction and task instance. For classification tasks, we convert the class labels as options into the instruction and ask the model to output the option index in natural language as the response.

Without any specific emphasis, we use the term "instruction" to refer to the combination of `Task_Instruction` and `Task_Instance`.

**Criteria for Task Selection.** To thoroughly investigate the issue of multi-task negative interference, we first established the Comprehensive Multimodal Task (CMT) benchmark, which is grounded in five key interference dimensions:

- **Interference of single and multiple image processing**: Visual Captioning and Visual Storytelling;

- **Interference between images with pure vision and multimodal information**: Text-Rich Images QA;

- **Interference between understanding global and local details**: Detailed Description and Spatial Inference;

- **Interference between visual and text recognition in images**: Spatial Inference and Knowledge OCR;

- **Interference between brief and detailed textual understanding**: Visual Captioning and Detailed Description. Building upon the aforementioned criteria, we can effectively and comprehensively benchmarking of diverse multimodal instruction capabilities across current MLLMs and our proposed HyperLLaVA.

**Task Collection and Categorization.** To comprehensively benchmark the diverse instruction following ability, we extensively gathered a wide variety of multimodal datasets from different fields and scenarios, and performed some processing to obtain the data we wanted, such as we used CogVLM Wang et al. (2023) to generate detailed descriptions for LAION-COCO and CC12M Changpinyo et al. (2021b). As illustrated in Figure 3, `CMT` has three important properties: **1) Demonstrative vision-language context,** all instructions contain sequences of (one or more) images and text that are highly correlated and together construct context, such as a storyboard with scripts, a textbook with diagrams. **2) Diverse forms of complex instructions,** the instructions range from designing panels for comics, to discovering differences between surveillance images, and to conversational embodied tasks. **3) Vast range of instruction-following scenarios,** the benchmark covers multiple practical scenarios, including cartoons, albums, *etc*.

**Evaluation Protocols.** Thanks to the unified task format of `CMT`, all tasks can be evaluated in a zero-shot manner. For the open-ended generation tasks, we adopt *ROUGE-L* for evaluation. For the tasks that require the models to output option indexes, we take *Accuracy* as the evaluation metric. While well-formated options are provided, we empirically observe that many MLLMs struggle to strictly follow instructions to output the option indexes but generate free-form text. Thus, when models do not exactly output the required options, we match their outputs to one of the given options.

**Benchmark Analysis.** Table 9 details the statistics. The CMT benchmark is divided into two parts: training and evaluation. `CMT` for training and `CMT` for evaluation both covers 7 tasks. In total, `CMT` for training includes 505,405 multi-round instruction-response pairs conversations and `CMT` for evaluation includes randomly selected 1,149 instruction-response pairs. On average, each instruction contains 1.53 images, 28.27 words and 1.74 images, 37.27 words, respectively.

**Table 10:** Summary of the instruction-following tasks in `CMT` benchmark.

| Task | Scenario | Dataset | Metric |
|---|---|---|---|
| **Visual QA** | | | |
| Visual Question Answer | Realistic Scene | VQAv2 Goyal et al. (2017a) | |
| Visual Question Answer with Reasoning | Realistic Scene | GQA Hudson & Manning (2019a) | |
| Visual Question Answer with External Knowledge | VQA with External Knowledge | OKVQA Marino et al. (2019) | |
| Ambiguous Visual Question Answer with Knowledge | Ambiguous VQA with Knowledge | AOKVQA Schwenk et al. (2022) | Accuracy |
| Visual Question Answer | Realistic Scene | ShareGPT ShareGPT (2023) | |
| Visual Question Answer | Non-Realistic Scene | JouneyDB Pan et al. (2023) | |
| **Visual Captioning** | | | |
| Text-Based Image Captioning | Non-Realistic Scene | TextCaps Sidorov et al. (2020) | ROUGE-L |
| Image Captioning | Non-Realistic Scene | JouneyDB Pan et al. (2023) | |
| **Spatial Inference** | | | |
| Visual Spatial Reasoning | Realistic Scene | RefCOCO Kazemzadeh et al. (2014) | IoU |
| Object Grounding | Realistic Scene | VG Krishna et al. (2017) | |
| **Detailed Description** | | | |
| Detailed Description | Realistic Scene | LAION-COCO Schuhmann et al. (2022) | ROUGE-L |
| Detailed Description | Realistic Scene | CC12M Changpinyo et al. (2021a) | |
| **Visual Storytelling** | | | |
| Animated Story Completion | Cartoon | AESOP (Ravi et al., 2021) | |
| Animated Story Completion | Cartoon | PororoSV (Li et al., 2019) | |
| Animated Story Completion | Cartoon | FlintstonesSV (Gupta et al., 2018) | ROUGE-L |
| Sequential Photo Storytelling | Album | VIST (Huang et al., 2016) | |
| Sequential Photo Storytelling | Cartoon | DiDeMoSV (Maharana et al., 2022) | |
| **Knowledge OCR** | | | |
| Knowledge OCR | Realistic Scene | LLaVAR Zhang et al. (2023) | ROUGE-L |
| Knowledge OCR | Realistic Scene | TextVQA Singh et al. (2019b) | |
| **Text-Rich Images QA** | | | |
| Slide QA | Slide | SlideVQA Tanaka et al. (2023) | |
| OCR QA | Book Cover | OCR-VQA Mishra et al. (2019) | Accuracy |
| Document QA | Document Image | DocVQA Mathew et al. (2021) | |

# 7 EXPERIMENTAL SETTINGS

## 7.1 IMPLEMENTATION DETAILS

In the training of the HyperLLaVA, we utilize the ADAMW Loshchilov & Hutter (2017) optimizer, adapting hyperparameters to cater to the specific requirements of each phase. For the feature alignment stage, parameters are set as $B = 32$, $Lr = 0.001$, while for visual instruction tuning stage, we adjust the parameters to $B = 16$, $Lr = 0.00002$. The configuration for the ADAMW optimizer incorporates the following settings: $\beta = (0.9, 0.999)$, $\varepsilon = 1 \times 10^{-8}$, and $W_d = 0.0$, ensuring a bespoke optimization strategy that effectively addresses the unique demands of each training phase.

Besides, We train our model following the same training process as LLaVA-1.5. The process includes two stages: (1) feature alignment stage: use 558K subset of the LAION-CC-SBU dataset to connect a frozen pretrained vision encoder to a frozen LLM; (2) visual instruction tuning stage: use a combination of 150K GPT-generated multimodal instruction-following data and approximately 515K VQA instances collected from academic-oriented tasks to guide the model in comprehending multimodal instructions. In addition to leveraging the identical training dataset as LLaVA-1.5, we introduce a supplementary `CMT` dataset comprising approximately 505K diverse data. This extensive dataset enriches the model's training regimen, bolstering its instruction-following performance and tackling complex visual tasks with greater finesse.

It is noteworthy that while LLaVA-1.5 accounts for the number of images in the input visual instruction task, it does not inherently possess the capability to comprehend intricate multi-image visual tasks. Instead, it confines responses to a single image, thereby forfeiting multi-image contextual information. HyperLLaVA extends this functionality by preserving all ¡image¿ tokens, sequentially substituting ¡image¿ tokens with image features, and employing corresponding masks to avoid loss impact. This augmentation enables the model to effectively process and respond to complex multi-picture visual task.

## 7.2 COMPARED METHODS

Recent advancements in LLMs (OpenAI, 2023a;b) have heralded significant achievements across various domains. Inspired by this success, many MLLMs (Li et al., 2023d; Liu et al., 2023b; Zhu et al., 2023; Alayrac et al., 2022a; Ye et al., 2023b; Gao et al., 2023; Li et al., 2023a) have been proposed to foster generalist vision-language reasoning. In our experiments, we conducted comparisons with some of the most recent and representative MLLMs in the following.

- **LLaVA-1.5** (Liu et al., 2023a) establishes a connection between the visual encoder ViT-L/14 from CLIP (Radford et al., 2021a) and the language decoder LLaMA (Touvron et al., 2023), utilizing a lightweight, Multilayer Perceptron (MLP) layer. Initially, the system trains this MLP layer using 558K image-text pairs, while keeping both the visual encoder and LLM static. Following this, LLaVA fine-tunes both the MLP layer and LLM using a dataset comprising 665K instructional vision-language pairs. The tested version are "LLaVA-1.5-7B" and "LLaVA-1.5-13B".

- **MiniGPT-4** (Zhu et al., 2023) bridges the gap between the visual encoder and text encoder using a fully-connected (FC) layer. Initially, this model trains the FC layer on a dataset comprised of 5M image-text pairs before fine-tuning it on 3.5K instructional vision-language data. Notwithstanding its simplicity, MiniGPT-4 requires the loading of a pre-trained vision encoder from BLIP2, as well as a Vicuna LLM (Chiang et al., 2023). The tested version is "minigpt4-aligned-with-vicuna7b".

- **BLIP2** (Li et al., 2023d) employs a dual-stage strategy to seamlessly bridge the modality gap, utilizing a lean Q-Former pre-trained on 129 million image-text pairs. The initial stage kick-starts the learning process of vision-language representation, leveraging a frozen image encoder, the ViT-g/14 from EVA-CLIP (Fang et al., 2023). Subsequently, the second stage harnesses a frozen LLM, the Vicuna (Chiang et al., 2023), to initiate the vision-to-language generative learning. This innovative strategy effectively facilitates zero-shot instructed image-to-text generation. The tested version is "blip2-pretrained-vicuna13b".

- **mPLUG-Owl** (Ye et al., 2023b) introduces a visual abstractor, fundamentally close the Perceiver Resampler in Flamingo (Alayrac et al., 2022a), as a bridge between the pre-trained visual encoder ViT-L/14 and the LLM (LLaMA (Touvron et al., 2023)). This model adopts a two-stage fine-tuning procedure. In the initial phase, both the visual encoder and the visual abstractor undergo comprehensive fine-tuning using a dataset of 204M image-text pairs. Subsequently, in the second phase, mPLUG-Owl applies the 158K LLaVA-Instruct dataset to fine-tune the pre-trained LLM in a parameter-efficient manner through the use of LoRA (Hu et al., 2021a). The tested version is "mplug-owl-llama-7b".

- **Otter** (Li et al., 2023a) is a multimodal model that applies in-context instruction tuning based on OpenFlamingo (Alayrac et al., 2022a). This model integrates a LLaMA-7B (Touvron et al., 2023) language encoder and a CLIP ViT-L/14. While the visual and text encoders remain static, Otter refines an additional 1.3 billion parameters. These parameters are derived from adaptation modules and are trained using 158K instruction-following data. The tested version is "OTTER-Image-LLaMA7B-LA-InContext".

- **InstructBLIP** (Dai et al., 2023a) originates from a pre-trained BLIP-2 model, which consists of a ViT-g/14 image encoder, a Vicuna LLM, and a Q-Former to act as the bridge between these two components. During the process of vision-language instruction tuning, only the Q-Former undergoes fine-tuning, with the training process leveraging data from 13 distinct visual question-answering datasets. The tested version is "blip2-instruct-vicuna7b" and "blip2-instruct-vicuna13b".

- **Shikra** (Chen et al., 2023) utilizes CLIP ViT-L/14 as the visual encoder and Vicuna as LLM, with a single fully-connected layer connecting the feature spaces of visual encoder and LLM. In both stages, freeze the visual encoder and tune all parameters in LLM. The model is trained in two stages, and freeze the visual encoder and tune all parameters in LLM in both stages. Shikra is able to comprehend user input of Points/Boxes and support the output of Points/Boxes, enabling seamless referential dialogue with humans. The tested version is "shikra-vicuna13b".

- **IDEFICS** (Laurençon et al., 2023) is an open copy of Flamingo, built on LLaMA and OpenCLIP (Ilharco et al., 2021). In the initial phase, OBELICS, a dataset containing 353

million images, was used for training. Subsequently, instruction fine-tuning was performed on 1 million data. The tested version are "idefics-9b-instruct" and "idefics-80b-instruct".

- **Qwen-VL** (Bai et al., 2023) utilized Qwen-7B as the LLM, Openclip's ViT-bigG as the vision encoder, and a single-layer cross-attention as the Vision-language adapter. A three-stage paradigm is used for training. In the first phase of pre-training on 1.4 billion data, freeze the large language model and only optimize the vision encoder and VL adapter in this stage. The second stage is multi-task pre-training, unlocked the large language model and trained the whole model at this stage. In the last stage, the Qwen-VL pre-training model is fine-tuned, freeze the visual encoder and optimize the language model and adapter module, and the interactive QWEN-VL-Chat model is generated. The tested version are "Qwen-VL-vicuna7b" and "Qwen-VL-Chat-vicuna7b".

## 8 ADDITIONAL EXPERIMENTAL RESULTS

We conducted additional experiments to further verify the strength of HyperLLaVA.

**Parameter-Efficient Fine-tuning.** Our proposed language expert also can serve as a parameter-efficient fine-tuning function. The structure is similar to the HyperNetwork+Adapter. However, original hypernetwork-based approaches generally condition their parameters on a learned latent embedding, implying the model is the same for every example, yielding performance decay. Summing up, the proposed language expert is an effective and parameter-efficient way to share information across multiple adapters to enable positive transfer to low-resource and related tasks.

**Detailed Performance on MME.** We report the detailed performance on the 14 subtasks of the MME benchmark in Table 14. MME benchmark measures both perception and cognition abilities on a total of 14 subtasks. We almost obtained the best score on each subtask compared to LLaVA 1.5, which further indicates the effectiveness of our method for diverse multimodal instruction understanding.

**Adaptation to other MLLM.** To study the generalizability of dynamic tuning to other MLLMs, we utilized our expert module to train MiniGPT-4. The outcomes of the vision-language tasks, as presented in Table 15, employing MiniGPT-4, are as follows. Our approach seamlessly integrates with MiniGPT-4, enabling it to proficiently tackle advanced vision-language tasks. For example, in the case of memes, MiniGPT-4 with the expert module accurately deciphers the complex humor in 11 out of 25 instances. In comparison to the original MiniGPT-4, the expert module yields a significant enhancement across all tasks, improving by 7 points for MiniGPT-4. These findings suggest that other baseline models equip the expert module can boost the capability for multi-modal tasks.

**Efficiency Comparsion.** Table 16 reports the comparison of model parameter counts and training time between HyperLLaVA and LLaVA. Notably, the parameters of the two models are similar in quantity, both the 7B and 13B versions. However, our HyperLLaVA achieves faster convergence in training time for the 7B version and comparable convergence training time for the 13B version, suggesting improved training efficiency for following diverse and complex multimodal instructions. We have not reported inference time, as the MLLMs produce outputs of varying lengths due to differences in instruction understanding.

**Qualitative Examples.** We show the qualitative examples generated by our HyperLLaVA in proposed CMT, including Detailed Description (Figure 7), Visual QA (Figure 8), Knowledge OCR (Figure 9), Visual Captioning (Figure 10), Visual Storytelling (Figure 11), Spatial Inference (Figure 12) and Text-Rich Images QA (Figure 13).

**Table 11: Comparison with LLaVA-1.5 (7B) and HyperLLaVA (7B) on MMMU benchmark Yue et al. (2024).**

| Methods | Art & Design | Business | Science-W | Health & Medicine | Human. & Social Sci. | Tech & Eng |
|---------|--------------|----------|-----------|-------------------|----------------------|------------|
| LLaVA-1.5-7B | 46.7 | 27.3 | 27.7 | 32.3 | 43.6 | 31.0 |
| HyperLLaVA-7B | **48.8** | **27.9** | **27.9** | **34.2** | **46.1** | **32.5** |

---

**Algorithm 1:** Vision-Language Alignment Framework

---

**Input:** Raw images $x$ and raw texts $T_r$ from PT datasets; Pre-trained models $\mathcal{M}_v(\cdot; \Theta_v)$ and $\mathcal{M}_l(\cdot; \Theta_l)$ with parameters $\Theta_v$ and $\Theta_l$ respectively;
**Output:** Projector Model $\mathcal{M}_p(\cdot; \Theta_p)$;

1   **Initialization:** Randomly initialize the parameters $\Theta_p$, including the visual HyperNetwork $\mathcal{H}_v$ and a 2-layer MLP; Freeze $\mathcal{M}_v(\cdot; \Theta_v)$ and $\mathcal{M}_l(\cdot; \Theta_l)$;
2   **for** $i \leftarrow 1$ **to** *number of epochs* **do**
3      **repeat**
4         Randomly sample a mini-batch;
5         Process data in batches to obtain $x$ and $T_r$;
6         Obtain $\mathcal{G}_V$ from $x$ using $\mathcal{M}_v(\cdot; \Theta_v)$ with Eq. (3);
7         Obtain $\mathcal{H}_V$ using Eq. (4);
8         Merge $\mathcal{G}_V$ and $\mathcal{H}_V$ to obtain $\mathcal{E}_v(\cdot; W_V^u(\cdot; \mathcal{H}_V), W_V^d(\cdot; \mathcal{H}_V))$ with Eq. (5);
9         Obtain $\mathcal{V}$ by integrating the output of $\mathcal{E}_v$ and a 2-layer MLP;
10        Obtain $\mathcal{T}$ and $\mathcal{R}$ with a tokenizer;
11        Concatenate $\mathcal{V}$, $\mathcal{T}$, and $\mathcal{R}$ as input tokens of $\mathcal{M}_l$, obtain $\hat{\mathcal{R}}$ by forward propagation;
12        Calculate cross-entropy loss (CEL) between $\mathcal{R}$ and $\hat{\mathcal{R}}$;
13        Update parameters $\Theta_p$;
14      **until** *No redundant data*;
15   **end**
16   **return** $\mathcal{M}_p(\cdot; \Theta_p)$

---

**Algorithm 2:** Multimodal Instruction Tuning Framework

---

**Input:** Raw images $x$ and raw texts $T_r$ from instruct-FT datasets; Pre-trained models $\mathcal{M}_v(\cdot; \Theta_v)$ and $\mathcal{M}_l(\cdot; \Theta_l)$ with parameters $\Theta_v$ and $\Theta_l$ respectively; Pre-trained projector model $\mathcal{M}_p(\cdot; \Theta_p)$ with parameters $\Theta_p$ from **Algorithm 1**;
**Output:** Large Language Model $\mathcal{M}_l(\cdot; \Theta_l)$, Projector Model $\mathcal{M}_p(\cdot; \Theta_p)$;

1   **Initialization:** Randomly initialize the parameters $\Theta_{\mathcal{H}_L}, W^Q, W^K$ for Eq. (6); Randomly initialize the parameters $W, B$ for Eq. (4); Freeze $\mathcal{M}_v(\cdot; \Theta_v)$;
2   **for** $i \leftarrow 1$ **to** *number of epochs* **do**
3      **repeat**
4         Randomly sample a mini-batch;
5         Obtain visual tokens $\mathcal{V}$, textual tokens $\mathcal{T}$, and response tokens $\mathcal{R}$ using procedures outlined in **Algorithm 1**;
6         Obtain hidden state token $h$ from the $\frac{L}{2}$-th layer through forward propagation;
7         Generate $\mathcal{G}_L$ from $h$ using Eq. (3);
8         Obtain dynamic MLP matrix $K$ using Eq. (4);
9         Combine $\mathcal{G}_L$ and $K$ to obtain $\mathcal{E}_l(\cdot; W_L^u(\cdot; \boldsymbol{w}, \boldsymbol{b}), W_L^d(\cdot; \boldsymbol{w}, \boldsymbol{b}))$ with Eq. (5);
10        Generate query sub-prompt $\hat{\mathcal{Q}}$ and key sub-prompt $\hat{\mathcal{K}}$ using Eq. (6) with $W^Q$ and $W^K$;
11        Generate $\hat{\mathcal{R}}$ through forward propagation of the next $\frac{L}{2}$ layers with dynamic MSA module using Eq. (6) and dynamic FFN module using Eq. (7);
12        Calculate cross-entropy loss (CEL) between $\mathcal{R}$ and $\hat{\mathcal{R}}$;
13        Update parameters $\Theta_{\mathcal{H}_\mathcal{L}}, W^Q, W^K, W, B$ and $\Theta_p$;
14      **until** *No redundant data*;
15   **end**
16   **return** Fine-tuned model $\mathcal{M}_l(\cdot; \Theta_l), \mathcal{M}_p(\cdot; \Theta_p)$;

---

Table 12: Comparsion of parameter-efficient learning.

| Methods | VQA Datasets | | Benchmark Toolkits | | CMT Benchmark | |
|---|---|---|---|---|---|---|
| | VizWiz | SQA$^I$ | MMB | SEED | VQA | SI |
| LoRa Hu et al. (2021b) | 51.5 | 68.4 | 63.2 | 60.4 | 77.8 | 35.4 |
| Adapter Houlsby et al. (2019) | 51.0 | 67.8 | 63.6 | 61.3 | 76.6 | 35.0 |
| HyperNetwork+Adapter Mahabadi et al. (2021) | 45.1 | 53.8 | 51.3 | 49.3 | 68.0 | 28.3 |
| Language Expert | **51.6** | **71.0** | **65.5** | **61.0** | **79.0** | **36.9** |

Table 13: Comparison with LLaVA-1.6 variant and simple version of HyperLLaVA1.6.

| Method | Visual QA | Visual Captioning | Spatial Inference | Detailed Description | Visual Storytelling | Knowledge OCR | Text-Rich Images QA |
|---|---|---|---|---|---|---|---|
| LLaVA1.6-8B$^\dagger$ | 81.5 | 22.7 | 35.3 | 34.8 | 20.7 | 49.6 | 32.3 |
| HyperLLaVA1.6-8B$^\dagger$ | **83.1** | **23.3** | **37.5** | **35.2** | **22.9** | **50.6** | **33.1** |

Table 14: Detailed zero-shot performance on MME benchmark.

| | BLIP-2 | InstructBLIP | LA-V2 | LLaVA | MiniGPT-4 | mPLUG-Owl | Otter | VPG-C | LLaVA-1.5 | HyperLLaVA |
|---|---|---|---|---|---|---|---|---|---|---|
| Existence | 160.00 | 185.00 | 120.00 | 50.00 | 115.00 | 120.00 | 195.00 | 180.00 | 185.00 | **185.00** |
| Count | 135.00 | 143.33 | 50.00 | 50.00 | 123.33 | 88.33 | 50.00 | 96.67 | 155.00 | **165.00** |
| Position | 73.33 | 66.67 | 48.33 | 50.00 | 81.67 | 50.00 | 86.67 | 80.00 | 133.33 | **133.33** |
| Color | 148.33 | 153.33 | 75.00 | 55.00 | 110.00 | 55.00 | 113.33 | 116.67 | 170.00 | **180.00** |
| Poster | 141.84 | 123.81 | 99.66 | 50.00 | 55.78 | 136.05 | 138.78 | 147.28 | 160.54 | **159.18** |
| Celebrity | 105.59 | 101.18 | 86.18 | 48.82 | 65.29 | 100.29 | 172.65 | 164.12 | 152.94 | **168.53** |
| Scene | 145.25 | 153.00 | 148.50 | 50.00 | 95.75 | 135.50 | 158.75 | 156.00 | 161.25 | **161.25** |
| Landmark | 138.00 | 79.75 | 150.25 | 50.00 | 69.00 | 159.25 | 137.25 | 145.00 | 170.50 | **172.25** |
| Artwork | 136.50 | 134.25 | 69.75 | 49.00 | 55.75 | 96.25 | 129.00 | 113.50 | 117.75 | **127.50** |
| OCR | 110.00 | 72.50 | 125.00 | 50.00 | 95.00 | 65.00 | 72.50 | 100.00 | 125.00 | **140.00** |
| Perception | 1293.84 | 1212.82 | 972.67 | 502.82 | 866.57 | 967.34 | 1292.26 | 1299.24 | 1531.31 | **1592.05** |
| Commonsense | 110.00 | 129.29 | 81.43 | 57.14 | 72.14 | 78.57 | 106.43 | 98.57 | 127.86 | **133.57** |
| Numerical | 40.00 | 40.00 | 62.50 | 50.00 | 55.00 | 60.00 | 72.50 | 77.50 | 42.50 | **60.00** |
| Text Translation | 65.00 | 65.00 | 50.00 | 57.50 | 55.00 | 80.00 | 57.50 | 57.50 | 77.50 | **65.00** |
| Code Reasoning | 75.00 | 57.50 | 55.00 | 50.00 | 110.00 | 57.50 | 70.00 | 87.50 | 47.50 | **75.00** |
| Cognition | 290.00 | 291.79 | 248.93 | 214.64 | 292.14 | 276.07 | 306.43 | 321.07 | 295.36 | **333.57** |

Table 15: Experiments of the experts for MiniGPT-4.

| Methods | Meme | Recipes | Ads | Poem | Total |
|---|---|---|---|---|---|
| MiniGPT-4 | 8/25 | 18/25 | 19/25 | 20/25 | 65/100 |
| MiniGPT-4+Expert | 11/25 | 20/25 | 19/25 | 22/25 | 72/100 |

Table 16: Comparison of model parameter counts and training time.

| Method | Params | Training Time |
|---|---|---|
| LLaVA-7B | 7,062,902,784 | ~18 hours on $8 \times A800$ |
| HyperLLaVA-7B | 7,192,424,080 (1.018 ×) | ~17.5 hours on $8 \times A800$(~0.972×) |
| LLaVA-13B | 13,350,839,296 | ~18.5 hours on $16 \times A800$ |
| HyperLLaVA-13B | 13,503,568,656 (1.011 ×) | ~18.5 hours on $16 \times A800$(~1×) |

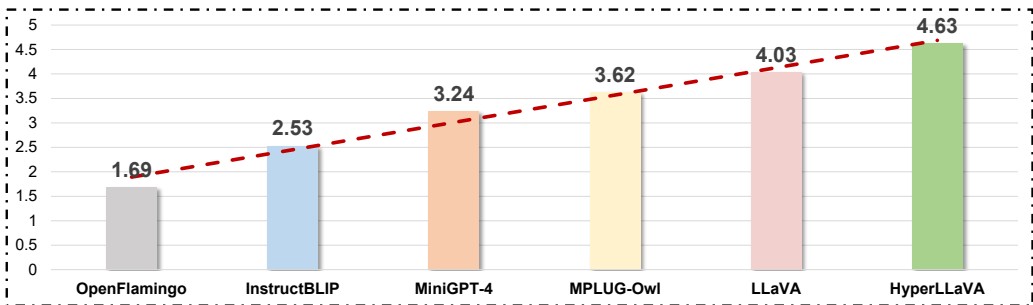

Figure 6: Human evaluation on OwlEval benchmark Ye et al. (2023b).

## 9 BROADER IMPACT AND LIMITATIONS

**Broader Impact.** The broader impact of HyperLLaVA, a general-purpose visual assistant, has potential benefits and risks associated with its deployment and release. The proposed HyperLLaVA serves as an upgrade version for LLaVA1.5, that enables dynamic projector learning and LLM tuning. By adaptively tuning both projector and LLM parameters, and integrating dynamical visual and language experts, we not only surpass the performance benchmarks set by LLaVA but also introduce a Comprehensive Multimodal Tasks (CMT) benchmark.

**Hallucination.** Similar to LLMs, HyperLLaVA might generate outputs that aren't grounded in facts or input data. This raises concerns about inferences made, especially in critical applications (*e.g.*, medical tasks).

**Bias.** Bias can be transferred from the base models to HyperLLaVA, both from the vision encoder (CLIP) and the language decoder (LLaMA/Vicuna). This may lead to biased outcomes or unfair representations of diverse content.

## Detailed Description

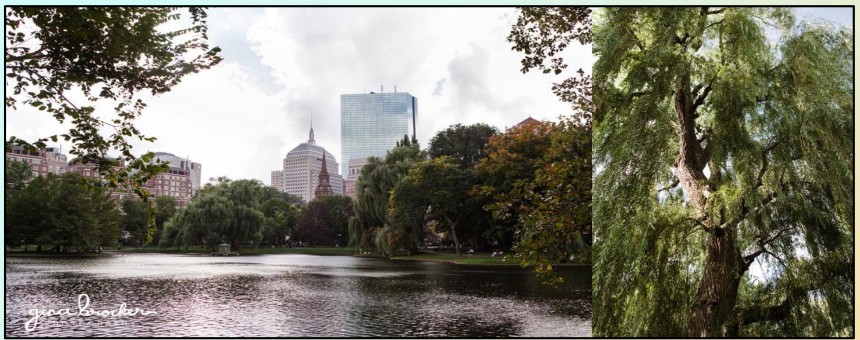

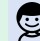 **Instruction**: Describe the image in detail.

The image captures a picturesque scene of a park with a large body of water, possibly a lake, surrounded by trees. The water is calm and reflects the surrounding greenery and the sky. The park is located in a city, as evidenced by the tall buildings visible in the background.

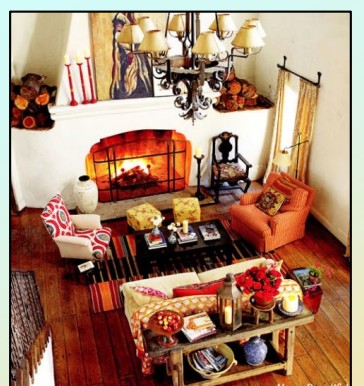

**Instruction**: Describe the image in detail.

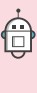 The image depicts a cozy living room with a fireplace as the centerpiece. The fireplace is lit, creating a warm and inviting atmosphere. The room is adorned with various pieces of furniture, including a couch, a chair, and a dining table. There are several books scattered throughout the room, indicating a love for reading. A vase is placed on the dining table, adding a touch of elegance to the space. The room is decorated with a mix of colors, including red, yellow, and orange, which contribute to the overall vibrant.

Figure 7: Qualitative examples in detailed description task.

Figure 8: Qualitative examples in Visual QA task.

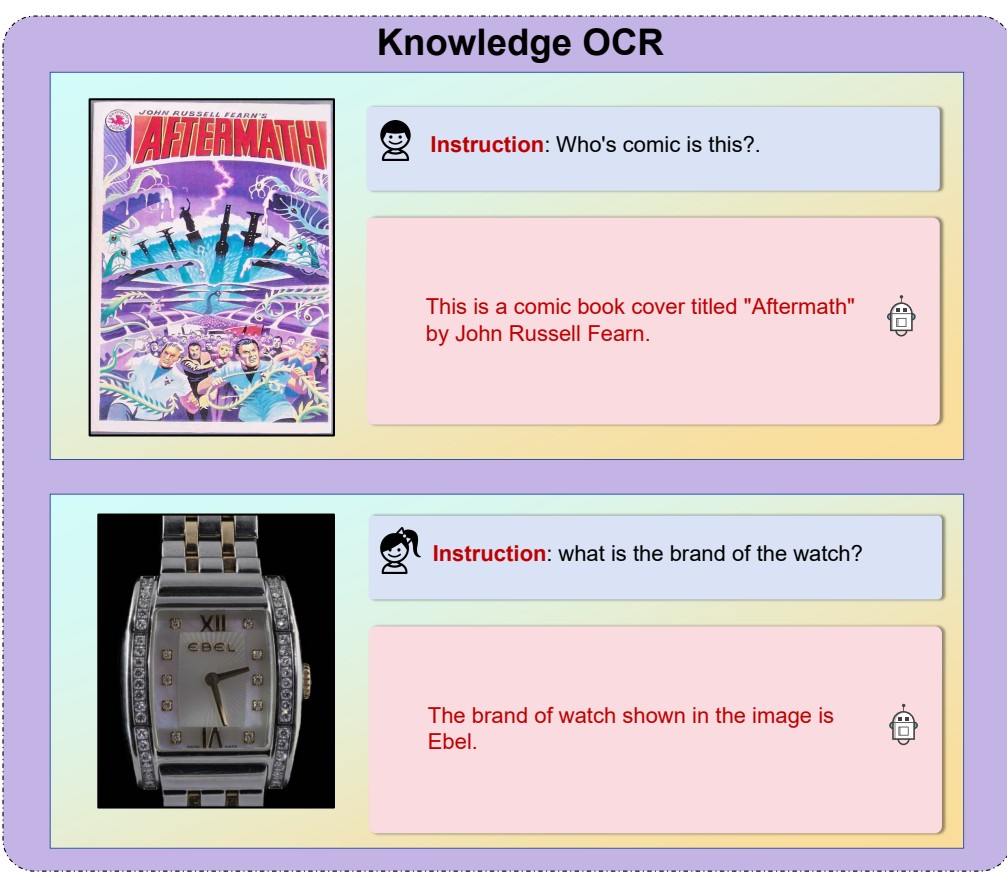

Figure 9: Qualitative examples in Knowledge OCR task.

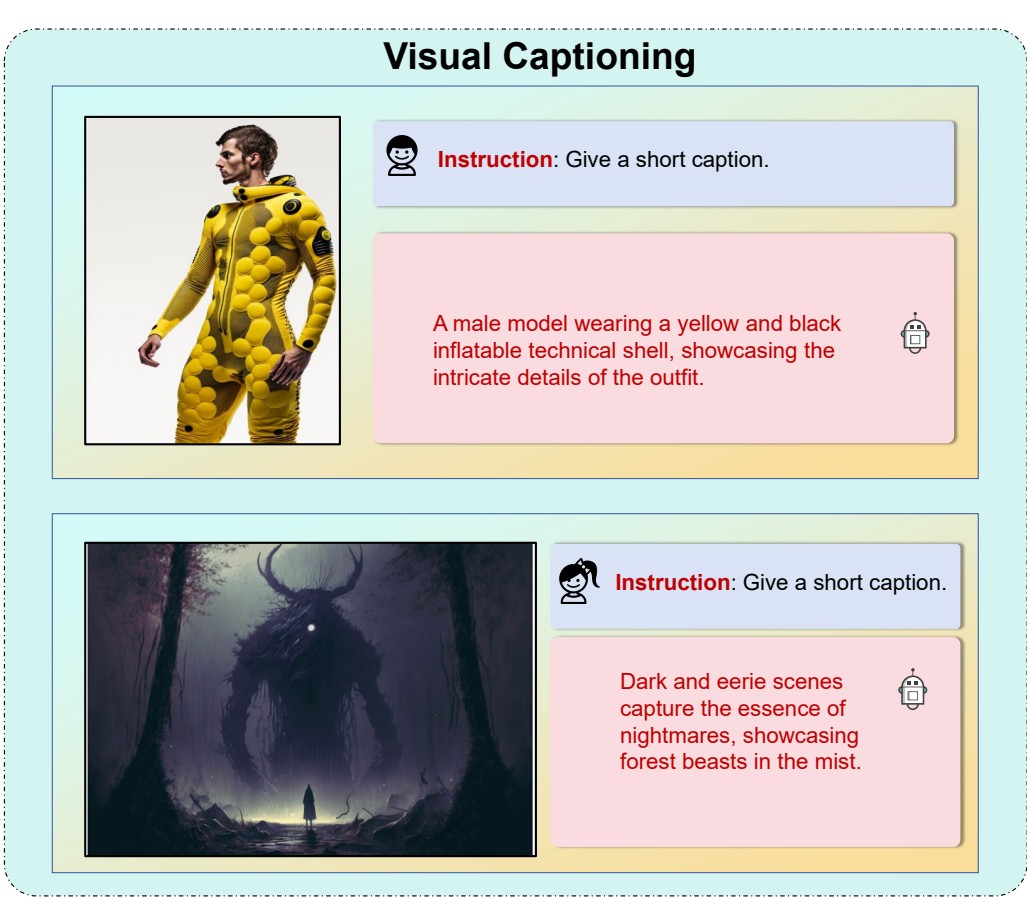

Figure 10: Qualitative examples in Visual Captioning task.

Figure 11: Qualitative examples in Visual Storytelling task.

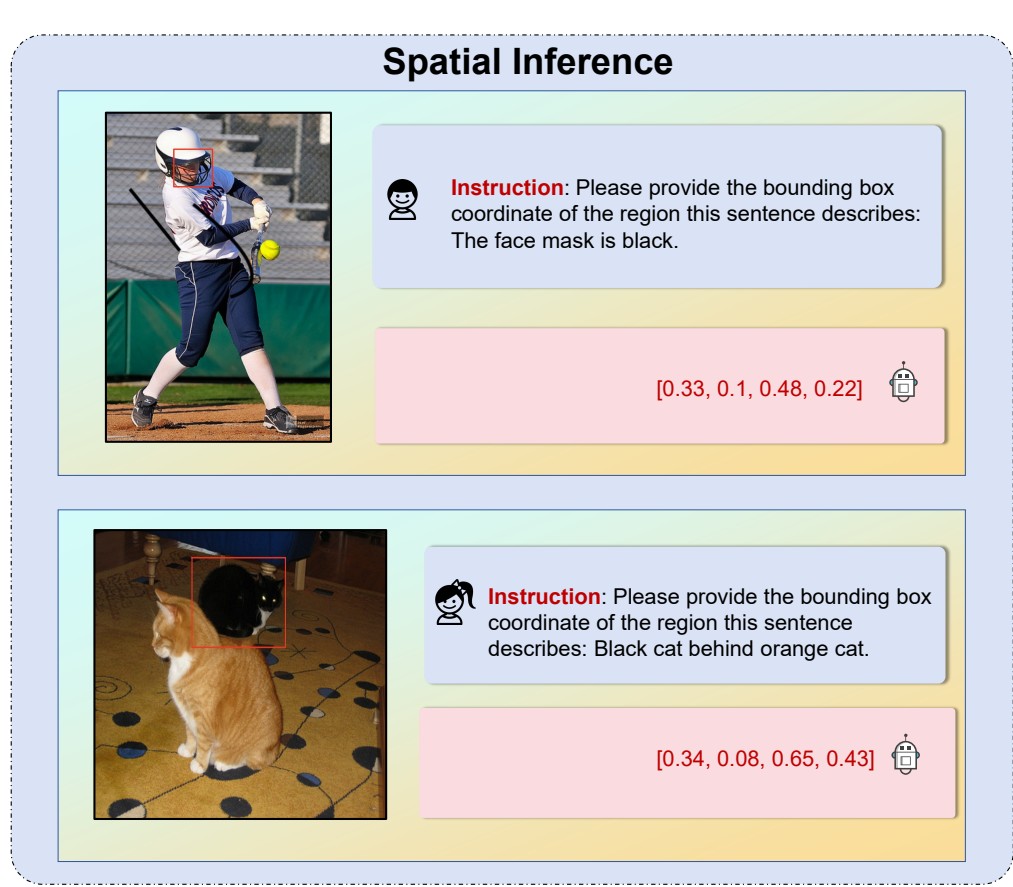

Figure 12: Qualitative examples in Spatial Inference task.

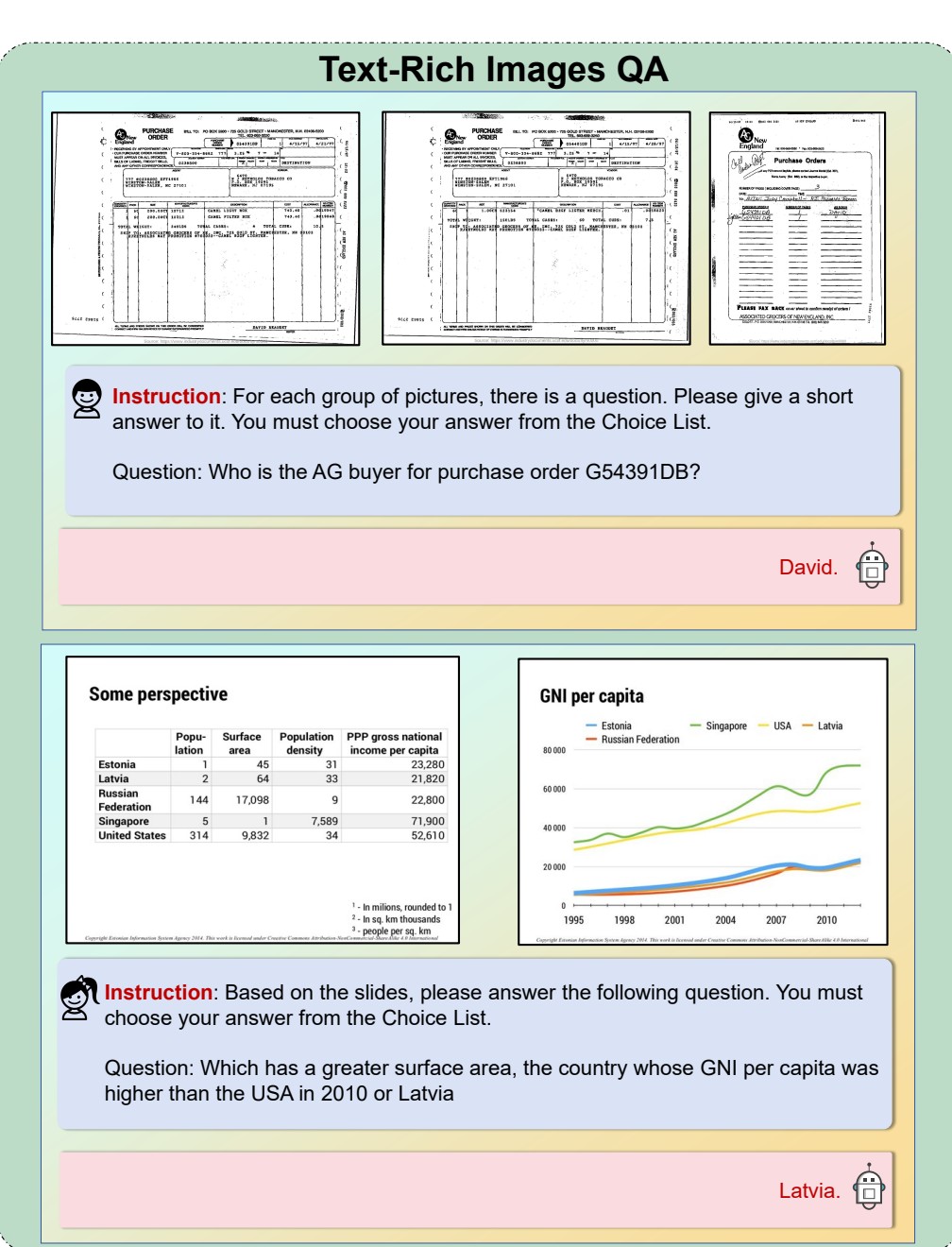

Figure 13: Qualitative examples in Text-Rich Images QA task.

