# OpenReview forum: "HyperLLaVA: Dynamic Visual and Language Expert Tuning for Multimodal Large Language Models"
_ICLR.cc/2025/Conference — ICLR 2025 Conference Withdrawn Submission_

### Official Review · Reviewer_H2so · 2024-10-30

**Soundness:** 1
**Presentation:** 2
**Contribution:** 1
**Rating:** 3
**Confidence:** 5

**Summary:**

This paper focus on the problem of the static mapping of existing MLLMs,  and propose a design of dynamic vision and language token experts to handle the claimed ``negative transfer'' of MLLMs. However, the reviewer think that the problem of this paper focuses is ill defined, which greatly contradicts the main principle of existing MLLMs. Meanwhile, the experimental results can not well support the argument of this paper as well as the design proposed in this paper.

**Strengths:**

Subjectively speaking, given an ill-defined problem, the other efforts made by the authors can hardly be regarded as the merits of this paper.

**Weaknesses:**

1. The main problem of this paper is its motivation. Task transfer learning is important for MLLMs, but it is hard to say negative transfer really matters in MLLMs. Above all, the most recognized principle of MLLMs is to use the giant LLM to learn massive multimodal data at the same time, thus generalizing to all downstream tasks. In this case, how can the authors argue that the static parameters are not good for large-scale MLLMs, just depending on the observation of [Want et al. 2019] experimented on the small-scale models and benchmarks.

More theoretical analyses and recent literature review are required to validate the motivation of this paper.

2. In addition, the design focus of this paper is also very problematic, i.e., learning dynamic projection for visual and text tokens. The main pricinple of most MLLMs is to project tokens of different modalities onto the same semantic, i.e. ,the one of LLM. Why the authors think the task-specific projection is better than this mainstream paradigm? Moreover, the text tokens have been well trained with the LLM architecture, thus why task-specific text embedding is needed?

Similarly, More theoretical analyses and recent literature review are required to validate this design .

3. The experiments do not support the argument of this paper. The overall improvement of the proposed Hyper-LLaVA is very marginal, which can be easily achieved by just increasing the training steps of LLaVA-1.5. Meanwhile, other experiments can also not strongly prove the motivation about the dynamic token design of this paper, as well as the qualitative analyses.

Some detailed suggestions:
1. Provide a direct comparison with LLaVA-1.5 trained for additional steps.
2. Include more rigorous statistical analyses  to demonstrate significance.
3. Design additional experiments that more directly test the benefits of the dynamic token approach.

Besides,  the complexity and additional steps of the proposed dynamic token projection are better to highlighted.

Overall, this paper states an argument against common research paradigm of existing MLLMs, but are not sufficiently supported in terms of either principle, methodology or experimental results.

**Questions:**

My concerns and questions are given in the weakness section.

---

### Official Review · Reviewer_dSaL · 2024-10-30

**Soundness:** 3
**Presentation:** 2
**Contribution:** 2
**Rating:** 5
**Confidence:** 4

**Summary:**

The static paradigm in current MLLM, like LLaVA, can introduce potential task interference, ie, the improvements for one task can cause performance degradation on other tasks.
Thus， the paper introduces HyperLLaVA to adjust the parameters of the vision projector and LLM conditioned on instruction semantics.

Experimental results are reported on commonly used benchmarks, like VQAv2, GQA, VizWiz, SQA, POPE, MME, MMB, SEED, and LLaVA-bench. Reasonable improvements are achieved.

**Strengths:**

(1) The method is simple to implement.
(2) Sound experimental results and ablations.

**Weaknesses:**

(1) The symbols are not consistent. What are the relationships between $\xi$, $E^{n}()$, and $H_M$ in Eqs (2), (3), (5)?

(2) What's the "L1" in line of 250?

(3) Why does the last input token $h^{L/2}$ at L/2-th layer can fully perceive the whole multimodal context?

(4) The comparison with LLaVA seems unfair because the paper uses an extra 505K CMT training data.
      For the HyperLLaVA, is LLM fully fine-tuned or trained with LoRA?

(5) Indicated by the formulation of $\mathcal{V}$ in line of 246 for vision projector and Eq. (6) for LLM, it seems not to be dynamic but an extra residual output.
Moreover, what's the network structure of $\xi$ in Eq. (6)?

**Questions:**

(1) The symbols should be defined systematically and consistently.
(2) The comparisons with baselines should be fair.

---

### Official Review · Reviewer_bQ3m · 2024-11-01

**Soundness:** 2
**Presentation:** 2
**Contribution:** 2
**Rating:** 3
**Confidence:** 5

**Summary:**

Recent advancements in scaling Multimodal Large Language Models (MLLMs) show improved performance on multimodal tasks, though static visual-language mappers like those in LLaVA limit potential across various tasks. To overcome this, This paper with HyperLLaVA introduces adaptive adjustments to both the projector and language model parameters through dynamic visual and language experts. These experts, generated by a hypernetwork, produce adaptive parameter offsets that respond to visual and language cues. This dynamic modeling is achieved through a two-stage training process. Experiments confirm that HyperLLaVA significantly outperforms LLaVA on multiple MLLM benchmarks, addressing limitations of static model adaptation.

**Strengths:**

The writing is fluent. The paper effectively identifies two challenges in applying hypernetworks to MLLM: weak correlation and unstable optimization.

**Weaknesses:**

The paper discusses the issue of unstable optimization in large language models and suggests a potential trade-off between the number of parameters generated by hyperparameters and optimization stability. However, the authors have only considered a minimal number of additional parameters (it would be helpful if the paper specifies what percentage these parameters represent of the original network). For instance, the addition of the visual expert is limited to the projector, which seems similar to adding a joint fine-tuning LoRA. The actual performance enhancement from the visual expert is unconvincing since it is only integrated at the input layer, resembling an enhanced connector, while many MLLM studies have shown that connectors contribute minimally to model performance.

Furthermore, in the language expert, the use of half-layer language guidance as input for the hypernetwork is quite unusual. Using the sample x itself, particularly the hidden layer embedding of its own sample, is rare (relevant literature would be appreciated if available). This self-generated network approach might suggest that it merely adds an extra structure rather than generating effective parameters, akin to LoRA, requiring many additional tuning parameters. It is unclear how the choice of using half of the layers was made. Will generating parameters for the second half of the network from the first half's output create a highly coupled network structure? Is there any pruning of structures like the hypernetwork during inference?

Lastly, the paper lacks clear motivation for the method used, and it is not clear what exactly constitutes the "unstable" performance mentioned in Table 5. Is this instability due to the simplicity of the projector structure or other optimization issues, such as drastic parameter changes? If adding experts to the projector laterally can improve these issues, why not utilize a multi-head approach in the projector? Considering inference costs in MoE and the unclear independence of the visual expert, these questions remain. Moreover, the experiments compare with LLaVA v1.5 NeurIPS 23, but it would be beneficial to compare with more recent works like CogVLM, Wings, and Mono InternVL. The performance improvements in Table 4 are almost all below 0.5%, which seems unconvincing.

**Questions:**

Is the training process of hypernetworks similar to the bilevel optimization seen in NAS (Neural Architecture Search)? What are the advantages and characteristics of parameters generated by hypernetworks compared to directly fine-tuned parameters? Could you possibly provide quantitative or visualized results?

In Table 5, the bolded section for MME accuracy seems incorrect. Typically, "LoRA" is the preferred capitalization rather than "LoRa." Additionally, the use of "L" in Equation 6 may lead to misunderstanding, like "Loss".

---

### Official Review · Reviewer_dfxV · 2024-11-02

**Soundness:** 2
**Presentation:** 2
**Contribution:** 2
**Rating:** 3
**Confidence:** 5

**Summary:**

The paper proposes HyperLLaVA, which generates dynamic parameter shifts through visual and language guidance, enabling
dynamic vision-language alignment and instruction tuning in two-stage training.  Experiments validate its effectiveness on some benchmarks.

**Strengths:**

1. The writing is good and easy for understanding.
2. Experiments are extensive to validate performance gains of the proposed method on some benchmarks.

**Weaknesses:**

1. MoE is not new in LLM and MLLM, which can also be a dynamic way for task transferring. Why not comparing with these methods, such as MoE-LLaVA, LLaVA-MoLE.
2.  Performance gains seem minor from experiments.  In Tab4, performance gains on SQA and MMB are almost ignoble.
3. Line 300-301, why authors claim that CMT is zero-shot, since it’s actually a supervised setting.
4. I also suspect the value of CMT benchmark, which is simply combined with existing datasets.  Actually, existing benchmarks like MMVet and MMBench contain more complex and diverse multi-modal instructions, which are strictly zero-shot.
5. All figures could be more clear for reading. It’s very difficult to understand the method from figures.

**Questions:**

See weakness

---

### Official Review · Reviewer_HzoZ · 2024-11-03

**Soundness:** 3
**Presentation:** 3
**Contribution:** 3
**Rating:** 5
**Confidence:** 4

**Summary:**

This paper addresses potential issues of task interference or negative transfer that may arise in the multi-task instruction tuning stage of MLLMs SFT training. Based on the concept of HyperNetworks, the authors propose dynamic visual and language experts that can adjust the corresponding projectors and LLM weights based on the semantics of the instruction. This approach aims to balance the potential negative effects of different tasks on each other during multi-task instruction tuning stage. Additionally, the authors introduce a new benchmark to comprehensively evaluate various aspects of MLLM capabilities. The effectiveness of their method is validated on both existing benchmarks and the new benchmark they proposed.

**Strengths:**

- This paper is clearly written and easy to follow
- This paper has conducted detailed experiments that show how each part works and validated that the overall method is effective.

**Weaknesses:**

- **Limited Scalability of the Method**: According to Table 1 in the paper, as the model size increases from 7B to 13B, the improvements provided by the proposed method over the baseline decrease, with most accuracy gains staying within a range of 0.1 to 1.0. This suggests that the method may lack scalability.
- **Limited Innovation of the CMT Benchmark**: The CMT Benchmark proposed by the authors seems to primarily integrate existing open-source benchmarks, lacking significant innovation compared to other established benchmarks. Additionally, the models tested on this benchmark are quite limited, and no evidence is provided to show that it offers any insightful conclusions.
- **More Ablation Study on Each Component**: Tables 3 and 4 focus on validating the effectiveness of the different experts proposed by the authors. In some benchmarks, such as SQA, SEED, and MMB, there are hardly any significant improvements. The authors seem to lack an ablation study to evaluate the effectiveness of combining these two modules. Additionally, Table 5 indicates that the performance improvements of the proposed method are quite limited.
- **Some Content Is Unclear**: The authors should specify the number of parameters introduced by their method. In a supervised training setting, it is difficult to determine whether the performance gains are due to the increase in parameters rather than the method itself.

**Questions:**

Please see the weaknesses above.

---

### Note · Authors · 2024-11-15

I have read and agree with the venue's withdrawal policy on behalf of myself and my co-authors.